# The Gelatin-Coated Nanostructured Lipid Carrier (NLC) Containing *Salvia officinalis* Extract: Optimization by Combined D-Optimal Design and Its Application to Improve the Quality Parameters of Beef Burger

**DOI:** 10.3390/foods12203737

**Published:** 2023-10-11

**Authors:** Maedeh Malekmohammadi, Babak Ghanbarzadeh, Shahram Hanifian, Hossein Samadi Kafil, Mehdi Gharekhani, Pasquale M. Falcone

**Affiliations:** 1Department of Food Science and Technology, Tabriz Branch, Islamic Azad University, Tabriz P.O. Box 11365-4435, Iranhanifian@iaut.ac.ir (S.H.); m.gharekhani@iaut.ac.ir (M.G.); 2Department of Food Science and Technology, Faculty of Agriculture, University of Tabriz, Tabriz P.O. Box 51666-16471, Iran; 3Drug Applied Research Center, Tabriz University of Medical Sciences, Tabriz P.O. Box 51656-65811, Iran; kafilhs@tbzmed.ac.ir; 4Department of Agricultural, Food and Environmental Sciences, University Polytechnical of Marche, Brecce Bianche 10, 60131 Ancona, Italy

**Keywords:** nanostructured lipid carrier, gelatin, sage extract, combined D optimal design, beef burger, shelf life

## Abstract

The current study aims to synthesize the gelatin-coated nanostructured lipid carrier (NLC) to encapsulate sage extract and use this nanoparticle to increase the quality parameters of beef burger samples. NLCs were prepared by formulation of gelatin (as surfactant and coating biopolymer), tallow oil (as solid lipid), rosemary essential oil (as liquid lipid), sage extract (as active material or encapsulant), polyglycerol ester and Tween 80 (as low-molecular emulsifier) through the high-shear homogenization–sonication method. The effects of gelatin concentrations and the solid/liquid ratio on the particle size, polydispersity index (PDI), and encapsulation efficiency (EE%) of sage extract-loaded NLCs were quantitatively investigated and optimized using a combined D-optimal design. Design expert software suggested the optimum formulation with a gelatin concentration of 0.1 g/g suspension and solid/liquid lipid ratio of 60/40 with a particle size of 100.4 nm, PDI of 0.36, and EE% 80%. The morphology, interactions, thermal properties, and crystallinity of obtained NLC formulations were investigated by TEM, FTIR, DSC, and XRD techniques. The optimum sage extract-loaded/gelatin-coated NLC showed significantly higher antioxidant activity than free extract after 30 days of storage. It also indicated a higher inhibitory effect against *E. coli* and *P. aeruginosa* than free form in MIC and MBC tests. The optimum sage extract-loaded/gelatin-coated NLC, more than free extract, increased the oxidation stability of the treated beef burger samples during 90 days of storage at 4 and −18 °C (verified by thiobarbituric acid and peroxide values tests). Incorporation of the optimum NLC to beef burgers also effectively decreased total counts of *mesophilic bacteria*, *psychotropic bacteria*, *S. aureus*, *coliform*, *E. coli*, molds, and yeasts of treated beef burger samples during 0, 3, and 7 days of storage in comparison to the control sample. These results suggested that the obtained sage extract-loaded NLC can be an effective preservative to extend the shelf life of beef burgers.

## 1. Introduction

Meat products, especially beef burgers, are one of the main food sources that, due to the presence of protein, fat, and water, act as an ideal environment for food pathogen growth, which leads to food spoilage and food-borne diseases. Microbial food spoilage is not only a risk factor for human health, but also a concern of great economic loss for the food industries involved in the production of packaged foods [1]. There are many nonthermal preservation methods for meat and its products, one of the simplest of which is the use of bioactive ingredients [1,2,3,4,5,6,7]. Sage (*Salvia officinalis* L.) is a medicinal plant of the *Lamiaceae* family, known for its interesting chemical properties for various biological activities such as antioxidant and antibacterial effects. The sage extract (SE) contains phenolic compounds, such as phenolic diterpenoids (carnosic acid, carnosol, rosmanol), phenolic acids (caffeic acid, rosmarinic acid, ferulic acid), and flavonoids (luteolin derivatives, apigenin derivatives) [8,9,10]. In contrast, the utilization of bioactive ingredients such as plant extracts in food products, is frequently limited due to low stability during food processing situations, which decreases their bioavailability as well as their functional characteristics [11,12].

Nanoencapsulation is a unique method of trapping bioactive compounds in a protective shell. This technique proves highly effective in stabilizing, transferring, solubilizing, preserving, and improving the activity of active ingredients [13,14]. In recent years, there has been significant attention in the food industry towards nanoencapsulation of bioactive ingredients using lipid-based systems and the production of nanocarriers for various applications [11,15,16]. Lipid-based carriers, mainly due to their low toxicity, low cost, and ease of production, are unique delivery systems that can load hydrophilic and lipophilic compounds, depending on the kind of lipids and microencapsulation method applied [17]. Nanostructured lipid carriers (NLCs), are one of these systems that are created of solid and liquid lipids, surfactants, and water, in which a liquid lipid core is surrounded by a matrix of solid lipids. NLCs can increase the stability of physicochemical properties, solubility, bioavailability, and controlled release of functional ingredients including nutraceuticals and natural preservatives in food products [18,19]. Therefore, NLC as a new carrier system can be very effective in both fields of producing enriched/functional foods and increasing the shelf life of foods by reducing their spoilage (food preservation). In recent years, various research has been carried out on the encapsulation of phytochemical compounds, extracts, and essential oils by NLC systems such as Mentha pulegium essential oil [20], saffron bioactive compounds [21], cardamom essential oil [22], lutein [23], Quercetin [24], olive leaf extract [25], cinnamon essential oil [26], camelina oil [27], and rosemary extract [28]. The use of polymer coatings for NLC can increase its effective performance, such as increasing the absorption rate and delivery efficiency, incrementing the chemical and enzymatic stability of the encapsulated material, and improving the colloidal stability [26]. Gelatin is a natural polymer that is obtained from part hydrolyzation of collagen in living organisms. This biopolymer was widely used in the food industry due to having unique features, including nontoxicity, inexpensiveness, availability, biocompatibility, and biodegradability [29]. In addition, several studies reported that gelatin can be easily applied as a thin coating to stabilize nanostructures owing to the presence of amine and carboxylic groups [2,30,31,32].

A combined D-optimal design is a type of experimental design that uses a computer algorithm to select the best set of treatment combinations from a candidate set, based on a specified model and an optimality criterion. The optimality criterion aims at maximizing the determinant of the information matrix, which results in minimizing the variance of the parameter estimates for the model. A combined D-optimal design can be useful when there are constraints on the number of runs or the design space, limited resources, a complex model that involves interactions or nonlinear terms, or when classical designs such as factorials and fractional factorials do not apply or are not feasible. A combined D-optimal design can be used for modeling complex processes and formulations by finding the optimal experimental settings and fitting a flexible and accurate model [33]. To the best of our knowledge, no study has been conducted on the encapsulation of plant extract as a natural preservative in the tallow oil-based/coated NLC system. Therefore, the current study aimed to optimize the formulation of the sage extract loaded in gelatin-coated NLC composed of tallow oil–rosemary essential oil with a D-optimal combined design. So, the main objectives were (1) preparation of SE loaded NLC using food-grade commercial lipids; (2) coating the obtained NLC using gelatin biopolymer and (3) using the optimum coated formulation for inhibiting pathogenic microbial and lipid oxidation in beef burger.

## 2. Materials and Methods

### 2.1. Materials

Fresh sage leaves were collected from the local market of Tabriz, Iran. Gelatin was obtained from Sigma Chemical Co., Ltd. (St. Louis, MO, USA). Tween-80, Polyglycerol ester (PGE), 1,1-diphenyl-2-picrylhydrazyl (DPPH), Dimethyl sulfoxide (DMSO), Chloroform, methanol, ethylene alcohol, and bacterial culture media were supplied by Merck Chemical Co. (Darmstadt, Germany). The desired bacteria were supplied by the Institute of Pasture (Tehran, Iran). Rosemary essential oil was supplied by Tabibdaru Company (Kashan, Iran). The beef tallow was obtained from a local slaughterhouse in Tabriz, Iran.

### 2.2. Sage Extract Preparation

In summary, 10 g of powdered sage leaves were added into 100 mL of hydro-alcoholic solution (70% ethanol) and stirred at room temperature for 72 h [34,35]. This solution was filtered through Whatman No. 1 filter paper and a rotary evaporator was used to remove the ethanol in this obtained supernatant (40 °C, 80 mbar, and 90 min). Then, the obtained final extract was freeze-dried at −40 °C with 0.3 hPa, increasing the temperature of the shelves from −40 °C to 17 °C in 24, and stored at −4 °C for subsequent usage [36].

### 2.3. Preparation of SE-Loaded Gelatin-Coated NLC

SE-loaded NLCs were prepared based on the high-shear homogenization method followed by sonication [26,37]. Briefly, tallow oil, as a solid lipid, and rosemary essential oil (REO) as a liquid lipid, were selected. Tallow oil was melted in the water bath at 70 °C. Then, REO was mixed with melted tallow oil, and polyglycerol ester was added to this lipid phase at a ratio of 1% *w*/*w*. An aqueous surfactant solution of Tween 80 was prepared as a dispersion medium (1% *w*/*w*) and warmed to 70 °C. In addition, gelatin powder at different concentrations (according to the table) was mixed separately with 12.5 mL of the aqueous phase to create a coating for NLC and was heated at 70 °C for 24 min. Finally, the prepared extract was added slowly into the lipid phase at 70 °C under a high shear homogenizer (Heidolph, Kelheim, Germany). In the next step, the aqueous surfactant solution was added, and then the aqueous phase containing gelatin was added drop by drop to the emulsion system. This system was homogenized at 20,000 rpm for 10 min. The synthesized NLC was sonicated with a probe sonicator (200 W, 24 kHz) (UP200H, Hielscher, Germany) (0.5 s cycle, 70% amplitude, and 10 min) and stored in the fridge.

### 2.4. NLC Analysis

#### 2.4.1. Particle Size, Polydispersity Index, and Zeta Potential

The particle size, polydispersity index (PDI), and zeta potential of stable NLCs and gelatin-coated NLC (GL-NLC) were evaluated using a dynamic light scattering (DLS), and Zetasizer (Zetasizer Nano-ZS, Malvern, UK) after 24 h of fabrication at 25 °C.

#### 2.4.2. Transmission Electron Microscopy

The morphology of the selected optimum NLC based on the obtained results of particle size, PDI, and EE investigations was characterized with Transmission electron microscopy (TEM) analysis using a Philips/FEI EM208S (Philips/FEI Co., Eindhoven, The Netherlands) [38].

#### 2.4.3. Encapsulation Efficiency (EE), and Stability (ES)

The encapsulation efficiency (EE%) and encapsulation stability (ES%) of SE loaded in NLC were investigated through the determination of the content of free and encapsulated SE [22]. Free and encapsulated SE were separated from each other using the ultrafiltration method (Millipore Amicon Ultra-15, MilliporeSigma, Burlington, MA, USA) by centrifugation. In brief, distilled water was added into NLCs dispersion (1:5) and centrifuged at 4000 rpm for 5 min (Universal 320, Sigma-Aldrich, Gillingham, UK). The absorbance of the obtained filtrate, which indicates the content of free SE, was measured at λmax = 325 nm through a UV–Vis Spectrophotometer (model Ultraspec 2000 Pharmacia Biotech, Markham, ON, Canada). The standard curve of SE was prepared using different concentrations of the extract (10–200 µg/mL) in water. (y = 0.0083x + 0.02.57, R^2^ = 0.9991). Finally, EE (%) was computed as follows [39]:$$\mathrm{EE}\;\left(\%\right)=\frac{\mathrm{content}\;\mathrm{of}\;\mathrm{encapsulated}\;\mathrm{SE}}{\mathrm{Total}\;\mathrm{SE}\;\mathrm{added}\;}\times 100$$

To investigate the stability of produced NLCs, they were placed in polyethylene microtubes and stored for 20, 40, and 60 days at 4 °C, and 25 °C, respectively, and the remaining percentage of encapsulated SE was measured as follows [40]:$$\mathrm{ES}\;\left(\%\right)=\frac{\mathrm{Encapsulated}\;\mathrm{SE}\;\mathrm{on}\;a\;\mathrm{specific}\;\mathrm{day}}{\mathrm{Encapsulated}\;\mathrm{SE}\;\mathrm{on}\;\mathrm{the}\;\mathrm{first}\;\mathrm{day}}\times 100$$

#### 2.4.4. Fourier Transform-Infrared Spectroscopy (FTIR)

FTIR measurements were performed through a spectrometer (Tensor27, Bruker Co., Ettlingen, Germany) using KBr pellets to distinguish the possible interactions that can occur during the synthesis of nanoparticles. This analysis was carried out in the range of 400–4000 cm^−1^ and with a resolution of 4 cm^−1^ [41].

#### 2.4.5. Thermal Analysis

The thermal performance was characterized through the differential scanning calorimetry (DSC) using a DSC thermal analyzer (LINSEIS. DSC model P 10). Four mg of lyophilized optimal GL-NLC with and without SE, pure SE, and pure NLC (without gelatin, and SE) was heated from 0 °C to 130 °C for 10 °C/min. An empty standard aluminum was used as a reference [42].

#### 2.4.6. X-ray Diffraction (XRD) Measurement

The lyophilized optimal GL-NLC with and without SE, pure SE, and pure NLC (without gelatin, and SE) were characterized by XRD analysis using an X-ray diffractometer (Tongda TD-3700, Dandong Tongda Science & Technology Co., Ltd., Dandong, China) using Cu Kα radiation with a wavelength of 0.1541 nm and a nickel monochromator filtering wave at 40 kV and 30 mA. The diffraction pattern was recorded at 2θ = 5–60° with a scanning speed of 0.04°/min at room temperature (25 ± 2 °C) [43].

#### 2.4.7. Antioxidant Activity 

The antioxidant activity of pure SE, and the produced NLC with and without SE, was calculated by measuring the scavenging of 2,2-diphenyl-1-picrylhydrazyl (DPPH) free radical assay. For this purpose, after a specific time (1, and 30 days), the NLC samples were centrifuged at 1000 rpm. Then, 2 mL of 0.1 mM of ethanolic DPPH solution was poured into 2 mL of the supernatant and pure SE (at different concentrations (50–1000 μg/μL). The reaction was performed at room temperature in a dark place for 60 min. Subsequently, with the help of UV–vis spectroscopy (Spectrum SP-UV500DB, Spectrum Instrumentation GmbH, Großhansdorf, Germany), the absorbance was determined at a wavelength of 517 nm [44,45]. The antioxidant activity (AA) of the samples was measured as Equation. The IC50 (μg/mL) value was also calculated.
$$\mathrm{Antioxidant}\;\mathrm{activity}\;\left(\%\right)=\frac{{\mathrm{Abs}}_{\mathrm{DPPH}}-{\mathrm{Abs}}_{\mathrm{sample}}}{{\mathrm{Abs}}_{\mathrm{DPPH}}}\times 100$$

#### 2.4.8. Microbiological Analysis

##### Bacterial Suspensions Preparation

*S. aureus* (ATCC25923), *P. aeruginosa* (ATCC27853), and *E. coli* (ATCC25922) were cultured on sterile Mueller-Hinton agar plates and incubated for 24 h at 37 °C. Then, the selection of a single colony for inoculation into a sterile tube with 2 mL of sterile normal saline to match the turbidity standard of 0.5 McFarland was performed. More dilution of the suspension in sterile normal saline to 3 × 10^5^ CFU/mL was performed before incubation.

##### Minimum Inhibitory Concentration (MIC) and Minimum Bactericidal Concentration (MBC)

MIC and MBC are defined as the lowest concentration of an antibacterial compound that is required to inhibit microbial growth and kill microorganisms, respectively [46]. Broth microdilution assay against *S. aureus*, *E. coli*, and *P. aeruginosa* was applied to measure the MIC and MBC of free SE, REO, SE-loaded NLC, and NLC without SE [22]. In summary, 100 μL of bacteria culture was poured into the wells of a 96-well microplate. Afterward, 100 μL of free SE, REO, SE-loaded NLC, and NLC without SE were added into the wells of the two columns, separately. 100 μL of each well was then transferred into the wells in the second column, and this process was repeated until all the wells of the column had half of the compound’s concentration of the previous column. Next, the samples were incubated at 37 °C for 24 h. In addition, each sample was recultured on Muller Hinton agar (MHA), and incubated for 24 h at 37 °C to investigate MBC. Finally, the growth or lack of growth of bacteria in different concentrations of the mentioned compounds was investigated, and MIC and MBC were determined. A positive control test (a culture medium containing bacteria without antibacterial compounds) and a negative control test (culture medium with antibacterial compounds and without bacteria) were conducted.

### 2.5. Preparation of Beef Burgers

To prepare a beef burger, frozen meat was defrosted and ground. The final product contained 95% meat, 1.5% salt, 2% spices, and 1.5% water. Certain amounts of free extract, SE-loaded NLC, and optimized NLC without SE were suspended in the water of formulation and immediately added to the meat. After adding all the ingredients to the minced meat (according to Table 1), stirring was performed manually for 2 min to distribute the ingredients evenly. Burgers weighing approximately 100 g were prepared by hand. Ascorbic acid was used as an antioxidant and sorbate as an antimicrobial compound in chemical and microbial tests. After preparation, the burgers were placed in sterile containers and stored at 4 °C for 7 days and −18 °C for 3 months. Microbial tests were performed at 0, 3, and 7 days at 4 °C. Chemical investigations were performed on 0, 7 days at 4 °C, and 90 days at −18 °C. The sensory evaluations were carried out on 0 and 90 days.

### 2.6. NLC-Enriched Beef Burger Analysis 

#### 2.6.1. pH Measurements

The pH of samples was determined using a Metrohm pH meter (827 pH Lab, Metrohm company, Herisau, Switzerland) after homogenizing 10 g of sample in 90 mL of distilled water for 10 s at 13,000 rpm with an ULTRA-TURRAXT25 (IKA, Königswinter, Germany). The measurements were performed in three repetitions after calibrating the pH meter with buffers 4 and 7 [47].

#### 2.6.2. Lipid Oxidation

Peroxide values (PV) and thiobarbituric acid reactive substances (TBARS) values were evaluated to determine primary and secondary lipid oxidation products, respectively. The TBAR measuring was performed using the method reported by [48], with minor modifications. In summary, 20 g of meat was added into 20 mL of trichloroacetic acid solution (10 *w*/*v* %). Then, it was homogenized by a homogenizer, and centrifuged. The obtained supernatant (2.0 mL) was added into the tube containing thiobarbituric acid (TBA) solution (0.01 M, 2.0 mL) and placed in a water bath (100 °C) for 20 min. The absorbance was read at 532 nm, and the content of malondialdehyde (MDA) (mg of MDA equivalent per kg of beef burger samples) was calculated by the following equation;
$$\mathrm{mg}\;\mathrm{MDA}/1000\;g\;\mathrm{sample}={\mathrm{Abs}}_{532}\times 5.4$$

To evaluate the peroxide value (PV) of the beef burger samples, the iodometric technique based on the titration of the samples with thiosulfate solution was carried out [48]. In summary, 1 g of the sample (oil extracted from the burger) was dissolved in a mixture of chloroform and acetic acid with a ratio of 3/2 *v*/*v* under magnetic agitation. Then 0.5 mL of saturated potassium iodide solution was added to this solution, and placed in the dark for 1 min. Then, 0.5 mL of starch reagent (1% *w*/*v*) was added and titrated with sodium thiosulfate (0.01 N). Finally, the PV (milliequivalent peroxide/kg sample) was calculated based on the following equation:$$\mathrm{PV}=\frac{1000\times S\;\times N}{W}$$
where S is the volume of consumed sodium thiosulfate in the titration, N is the normality of sodium thiosulfate, and W is the weight of oil extracted.

#### 2.6.3. Microbial Analysis

Microbial tests were performed on beef burger samples, including the total count of *mesophilic bacteria*, *psychrotrophic bacteria*, *S. aureus*, *coliform*, *E. coli*, *salmonella*, molds, and yeasts. In summary, 10 g of beef burger sample was mixed with 90 mL sterile peptone water (0.1%) in a stomacher (Bagmixer 400P, Interscience, France) for 2 min. After that, different serial dilutions (1:10) of microorganisms were prepared in sterile peptone water (0.1%) and inoculated in the plates that were previously incorporated with culture media. Then, the total mesophilic viable count (TMVC), and psychrotrophic bacteria were investigated on plate count agar (PCA) incubated at 30 °C for 48 h and 5 °C for 10 days, respectively [49]. Baird-Parker agar (BPA) supplemented with egg yolk tellurite was used to count total *S. aureus* after 48 h incubation at 37 °C [50]. Samples inoculated with *Salmonella* were plated on Xylose Lysine Deoxycholate agar (XLD) overlayed with Tetrathionate broth (TTB) using the thin-layer agar method to allow for the recovery of injured cells [50]. Enumeration of *E. coli* (EC) was performed with the most probable number (MPN) method as reported in the bacteriological analytical manual (BAM) [51]. The l mL of each dilution series was transferred in triplicates into 16 mm Durham tubes containing 9 mL of lauryl sulfate broth (LSB) and gently rotated to suspend any adhering matter into the liquid. All tubes were incubated at 37 °C for 48 h and observed for turbidity and gas production, which was indicative of the positive result. All tubes with positive results were subcultured in 16 mm Durham tubes of EC broth by transferring 1 mL inoculants into 9 mL broth. The tubes were further incubated at 44 °C for 24 h and observed for turbidity and gas production. The MPN was calculated based on the results of the EC tubes. The coliform bacteria counts were measured using violet red bile agar (VRBA) and incubated at 37 °C for 24 h. The yeast and mold counts were determined on Yeast Extract Glucose Chloramphenicol (YGC) medium at 25 °C for 5 days’ incubation [52]. Bacterial counts were reported as log CFU/g samples.

#### 2.6.4. Sensory Characteristics

Sensory evaluation of beef burger samples on 0 and 90 days was analyzed by 40 random untrained panelists. Raw burger samples were evaluated in terms of odor, color, and overall acceptability. Panelists were asked to evaluate odor, flavor, color, and overall acceptability using a 1–5 hedonic scale, with 1 corresponding to the least-liked beef burgers and 5 corresponding to the most-liked. A score of 3.5 was considered the limit of acceptance or rejection of the sample in terms of sensory evaluators. All samples were raw and there no need to be served; however, informed consent was obtained from the panelists [53]. 

### 2.7. Experimental Design and Optimization

In the present work, a combined D-optimal design was used to evaluate the effect of two independent preparation variables, including one mixture variable (i.e., solid lipid/liquid lipid ratio) and one numerical variable (i.e., gelatin protein concentration) on the dependent variables (responses) including particle size, polydispersity index, and encapsulation stability. In this regard, 14 runs were conducted to select the best synthesized NLC sample. Preparation variables and their values at coded levels are summarized in Table 2. All calculations and graphics were carried out using the statistical software Design Expert 7.0 (Version 7.0.2, State-Ease, Minneapolis, MN, USA) and Excel. The 3D surfaces and contour graphs were used to analyze the effects of variables on responses. The following polynomial equation was fitted to the data: $$Y=\beta_{0}+\sum_{i=1}^{k}\beta_{i}X_{i}+\sum_{i=1}^{k}\beta_{ii}X_{i}^{2}+\sum_{\begin{array}{c} i=1\\ i<j \end{array}}^{k-1}\sum_{j=2}^{k}\beta_{ij}X_{i}X_{j}$$
where *β*_0_, *Y*, and *k* correspond to a constant coefficient, a response variable, and the number of variables, respectively. *β_ii_*, *β_ij_*, and *β_i_* are the measures of the $X_{i}^{2}$, *X_i_X_j_*, and *X_i_* of quadratic, interaction, and linear effects, respectively. *X_j_* and *X_i_* are independent variables in coded units.

Several statistical parameters of various polynomial models consisting of the coefficient of determination (R^2^), the adjusted coefficient of determination (adjusted R^2^), the predicted coefficient of determination (predicted R^2^), lack of fit, and the coefficient of variation were performed to select the best-fitting models. 

Statistically significant differences between means were calculated by the one-way analysis of variance (ANOVA) and Duncan’s multiple range test at a significance level of 0.05 by SPSS software (Version 20, SPSS Inc., Chicago, IL, USA).

## 3. Result and Discussions

### 3.1. Combined D-Optimal Optimization 

A combined D-optimal design was used to select the best model and obtain the optimal formulation based on one mixture variable (i.e., solid lipid/liquid lipid ratio) and one numerical variable (i.e., gelatin protein concentration).

According to this experimental design, the effects of solid-to-liquid lipid ratio and concentrations of gelatin coating on dependent variables, including the particle size, polydispersity index (PDI), zeta potential (ZP), and encapsulation efficiency (EE%), of the NLC dispersions were performed, and the results are shown in Table 2 and Figure 1. The statistical parameters calculated by the design-expert software are listed in Table 3 and Table 4. As can be seen, the significance of the model, the nonsignificance of the lack of fit, the high coefficient of determination (R^2^ = 0.97), and the adequacy of accuracy (>4) indicate the validity and accuracy of the fitted model for predicting dependent variables. Particle size and Zeta potential were best-fitted using quadratic models, while the linear model was found to be the best-fitted model for PDI values (Table 3). The final fitted equations in terms of L_Pseudo components, coded process factor, and coded categoric factor with the significance coefficients were:*Y*_1_ = 90.56 *X*_3_ − 35.92 *X*^2^_3_ + 225.49 
*Y*_2_ = 0.40 *X*_1_ + 0.46 *X*_2_ + 0.042 *X*_1_*X*_3_ + 0.12 *X*_2_*X*_3_

*Y*_3_ = −7.12 *X*_1_ − 6.01 *X*_2_ + 10.36 *X*_1_*X*_3_ + 11.67 *X*_2_*X*_3_

*Y_4_* = 69.15 *X*_1_ + 74.03 *X*_2_ − 32.16 *X*_1_*X*_2_ + 5.00 *X*_1_*X*_3_ − 12.41 *X*_2_*X*_3_ − 13.50 *X*_1_*X*_2_*X*_3_ − 18.4 *X*_1_*X*^2^_3_ − 6.23 *X*_2_*X*^2^_3_ + 60.88 *X*_1_*X*_2_*X*^2^_3_

where *Y*_1_, *Y*_2_, *Y*_3_, and *Y*_4_ are particle size, PDI, Zeta potential, and encapsulation efficiency, respectively.

The concentration of gelatin (GL) had a significant linear and quadratic effect on the responses of particle size, PDI, and ZP responses. The results showed that the increment of GL concentration from 0.1 to 0.8 g/100 g suspension led to a significant increase in particle size (*p* < 0.05) at all solid/liquid lipid ratios (Figure 1a). The different NLC samples showed particle sizes ranging from 100 to 287 nm falling within the nano and submicron scale. The increase in the solid/liquid lipid ratio from 60/40 to 90/10 had no significant effect on particle size at all gelatin concentrations. Similarly, to mean particle size results, the PDI values increased from 0.36 to 0.6 with increasing gelatin concentrations from 0.1 to 0.8 g/100 g suspension at all solid/liquid lipid ratios; however, it was more significant at low solid/liquid lipid ratio (60/40) (Figure 1b). PDI is an indicator of the uniformity of particle size distribution in the colloidal systems. It ranges from 0.1 to 1 in real colloidal systems and the high value of it shows wider and more nonuniform particle size distribution. Several parameters can affect the particle size of NLC particles such as type of lipid ingredients, solid/liquid lipid ratio, type and concentration of emulsifier and active encapsulated material, homogenization and sonication variables, and cooling rate of pre-emulsion. The low-molecular weight surfactants at optimum concentrations usually can decrease the size of final NLC nanoparticles by decreasing interfacial tension and increasing the pre-emulsion stability, preventing from recrystallization of lipid crystals from beta prim to beta form and inhibiting the aggregation of lipid nanocrystals. However, in current research, gelatin proteins acting as a high molecular weight surfactant increased particle size, which could be attributed to the formation of a thick protein layer in the interface of pre-emulsion (unlike low-molecular surfactant) or change in surface electric repulsion, which in turn increase droplet diameters. Ref. [54] reported particle size of 71.4 to 366.3 nm and PDI value from 0.14 to 0.36 for the pomegranate seed oil-loaded NLCs, which remained stable during 2 months of storage. They declared the particle size and PDI reduction by increment of emulsifier percentage from 2 to 6. Ref. [55] declared that the preparation of NLCs with long-chain triacylglycerols led to high viscosity during the hot homogenization method and, consequently, increased the mean particle size of the produced particles. On the other hand, the fats composed of shorter-chain fatty acids have lower interfacial tension than those containing longer-chain fatty acids, leading to smaller droplet size in the emulsion. Ref. [56] reported that an increment in homogenization cycles did not significantly affect the particle size of phytosterol-loaded NLC but reduced the PDI. Particle size and PDI parameters also stayed stable after 15 days of storage at 25 °C. In a similar work, ref. [57] utilized an NLC system to encapsulate Hibiscus sabdariffa extract. The particle size was reported as 470 nm for microwave-assisted extract-loaded NLC and 344 nm for pressurized liquid extract-loaded NLC. In another research work, [20] studied the encapsulation of Mentha pulegium essential oil into NLC and reported particle sizes of 40 to 250 nm and PDI of 0.4 for their NLC samples.

The zeta potential value shows the amount of electrostatic repulsion between adjacent, similarly charged droplets and particles in a colloidal system. In the current research, the zeta potential of NLCs changed from negative (~−18.4 mv) to positive values (~+8 mv) with increasing the gelatin concentrations from 0.1 to 0.8 at all solid/liquid lipid ratios (Figure 1c). This change may be responsible for the increase in particle size due to increased flocculation phenomena. Increasing solid/liquid lipid ratios had no significant effect on zeta potential at all gelatin concentrations. Tween 80 is a nonionic surfactant, and the negatively charged surface of NLC was probably related to the free hydroxyl groups of phenolic compounds in SE extract and essential oils or free fatty acid (carboxylic acid) and the phosphate group of phospholipids, which naturally present in most edible oils and lipids. The surface electrical charge of particles and droplets is considered a key indicator of their colloidal dispersion stability. Thus, it can be assumed that higher electric charges on the surface can lead to higher repellent forces among particles and droplets, which in turn can prevent aggregation. When the zeta potential is small, attractive forces may exceed this repulsion, implying the system may become unstable and particles join together (aggregation/flocculation/coalescence). The electrostatically stabilized dispersions usually need a zeta potential greater than ±30 mV to have good long-term stability over storage time; however for colloidal systems stabilized with a combination of ionic and nonionic emulsifiers, a zeta potential of ~±20 mV is sufficient for particles to repel one another due to the presence of steric hindrance resulting from large molecules and long chain hydrocarbons of nonionic emulsifier. Ref. [54] prepared pomegranate seed oil-loaded NLCs, which were stable during 2 months of storage. They reported zeta potential values from −18.3 to −27.2 mV, which could contribute to long-term physical stability. They also declared that a higher surfactant concentration reduced the zeta potential of NLC formulations. Ref. [18] developed a turmeric extract-loaded nanostructured lipid carrier. They showed that the optimum formulation had an average particle size of 112.4 nm, PDI of 0.37, and zeta potential of −23.3 mV. 

The encapsulation efficiency of NLC samples (EE%) did not follow a specific pattern and ranged from 45 to 80%. Maximum EE% has been observed at a 0.1% gelatin and 60/40 solid-to-liquid lipid ratio. It depended more on the solid/liquid lipid ratio than gelatin concentration. According to previous studies, EE% is dependent on the solid/liquid lipid ratio, type and concentration of surfactants, the lipophilicity of encapsulant (solubility of the bioactive in the NLC matrix), and preparation method of NLC formulation [58]. Ref. [54] reported EE% of 96–99% for the pomegranate seed oil-bearing NLC formulations, with a high amount of EE% related to the heterogeneous composition of different lipids. Ref. [57] encapsulated Hibiscus sabdariffa extract into NLC. They obtained different EE% of quercetin and anthocyanins for extracts prepared by microwaving and pressurizing methods.

To overall optimize the formulation of NLCs, the goals of achieving low particle size, high zeta potential, and low PDI were considered and the obtained results were evaluated with the desirability function, where desirability higher than 0.7 indicates optimal conditions. The highest desirability was related to the sample with a concentration of 40% liquid lipid (REO), 60% solid lipid (tallow oil), and a GL concentration of 0.1 g/100 g with desirability of 0.97. The values of particle size (nm), zeta potential (mV), and PDI for the selected optimal sample were obtained as 100.4 nm, −18.4 mV, and 0.36, respectively.

To assess the validity of the regression models, the actual data were compared with the values predicted by the models. The percentage error of theoretical value and actual value for particle size, PDI, ZP, and EE was 6.34, 5.55, −4.43, and −1.53, respectively. 

The ES% of the optimum SE-loaded GL-coated NLC was measured during 60 days of storage at 4 °C and 25 °C (Figure 2). As can be seen, immediately after production, the ES% level was the same in both temperatures, while over time, the level of ES% significantly decreased and this decrease was greater at 25 °C than at 4 °C. 

### 3.2. Morphological Studies with TEM

The size and morphology of the optimized NLC sample were investigated using the transmission electron microscopy (TEM) technique (Figure 3I). The result obtained from TEM analysis confirmed the results of particle size analysis obtained by the DLS technique (Figure 3II,III). Both methods showed that the particle size was in the range of ~100 nm. Moreover, the TEM image showed that the optimal NLCs had a spherical shape and a narrow size distribution. The shape of lipid-based nanoparticles is influenced by the rate of lipid crystallization, the lipidic phase ingredients, and the shape of lipid crystals. It was reported that an ordered crystalline structure (β modification) leads to elongated crystals and the spherical shape of crystals can be regarded as a confirmation of a less ordered crystalline structure (amorphous structure) of lipidic nanoparticles [54]. Ref. [27] confirmed the spherical and smooth surface of the optimum camelina oil containing NLC formulations. Ref. [39] used the NLC system to encapsulate hesperetin and utilized TEM to study the morphology of samples. This study confirmed the spherical particle shape and confirmed the results of particle size analysis. In another research, [57] confirmed the spherical shape and narrow particle size distribution of the *Hibiscus sabdariffa* extract-loaded NLC via TEM.

### 3.3. FTIR Analysis

FTIR analysis is an excellent technique to study the possible interactions between the compounds used in the produced NLC structure. The changes in the structure or creation of new compounds can be identified by displacement of bands, variation of band length, or exhibition of new bands.

The FTIR results are presented in Figure 4. The SE spectrum displays the specific bands at 3422 cm^−1^, attributing to hydroxyl groups (OH) of phenols, 1701 cm^−1^ corresponding to the carbonyl group (C=O) of the esters and the carboxylic acid in salvianolic acid and rosemarinic acid, 1405–1610 cm^−1^ (sp^2^ carbons of the aromatic ring), and 1038–1361 cm^−1^, referring to stretching vibrations of the C–O bonds in alcohols and carboxylic acids [59]. The FTIR of GL demonstrated the specific bands at 3435 cm^−1^, 1657 cm^−1^, 1550 cm^−1^, and 1220 cm^−1^, which could be attributed to the amide A, amide I (C=O stretching), amide II (N-H bending,) and amide III (C-N stretching), respectively [60]. Observing the bands at ~1159, 1744 cm^−1^ (vibration bands related to beef tallow triglyceride) [61], 1374, and 1214 cm^−1^ (vibration of CH_3_(CO), C-O-C and CH_2_ groups in REO) [62], in the spectrum of the produced NLC, indicating the successful production of the combined NLC. The shifts of band positions in the FTIR spectrum of produced NLC containing SE and coating with GL (optimum NLC), indicate that SE has been successfully loaded in the produced NLC, and well coated with GL. Ref. [27] confirmed no interaction between ascorbyl palmitate and the NLC (made with camelina oil) system by FTIR analysis. Ref. [25] used FTIR spectroscopy to show the successful incorporation of Spanish olive leaf extract (oleuropein) into nanostructured lipid carriers. Ref. [39] utilized the FTIR spectrum to analyze the functional groups of pure materials and the hesperetin-loaded NLCs. They showed that the main IR bonds of pure materials were present in the FTIR spectrum of loaded NLCs but were shifted toward higher wavelengths. Ref. [18] studied the FTIR spectrum of the turmeric extract NLC. They declared that major bands of pure turmeric extract disappeared in turmeric NLC, indicating the successful encapsulation of turmeric extract into NLC.

### 3.4. Thermal Analysis by DSC

The type and amount of solid and liquid lipids, surfactants, and active materials can affect the thermal behavior of NLC. Differential scanning calorimetry (DSC) is used to evaluate the melting and crystallization behavior of lipid-based materials because various lipid structures have different thermal properties. The evaluation of the crystalline and polymorphism behavior of the lipid carrier and the effect of SE encapsulation on the melting behavior of the produced nanocarrier system was performed with the DSC method. Figure 5 shows the thermogram of the free SE extract, NLC without sage extract, NLC without gelatin coating, and optimal SE-loaded GL-coated NLC. According to Figure 5, free sage extract powder did not show any sharp peaks, which could be attributed to the amorphous nature of the extract powder and the lack of melting phenomena [59]. In the DSC curves of NLC samples, the endometrial melting peak could be observed at a temperature lower than 50 °C, which indicates the presence of irregular crystalline phases of lipids [63]. Considering the use of a combination of solid and liquid lipids in the structure of NLC, this behavior was expected. The presence of different triglycerides and essential oils in NLC compositions can lead to less order and an incomplete lipid crystalline structure with a low melting point, which in turn potentially helps increase encapsulation efficiency and stability during storage. The addition of the sage extract to the GL-coated NLC did not change the peak intensity (sharpness); however, the temperature and width of the peak changed slightly, which could be attributed to the incorporation of sage extract into the NLC network and the decreased crystal order and compactness. Incorporation of bioactive core material in the NLC structure can be confirmed by decreasing the peak intensities and melting temperature and also widening the peak as a result of the increase in the irregularities in lipid crystals. By comparing the thermogram of the uncoated NLC and GL-coated NLC, it could be concluded that gelatin increased the crystalline structure and peak intensity in the GL-coated NLC [64]. Gelatin is a semicrystalline material and usually shows endothermic peaks between 75 and 95 °C relating to melting and protein denaturation. Ref. [56] developed an NLC system to encapsulate free phytosterol. Fully hydrogenated soybean oil (FSHO) showed only one peak at 47 °C, which is related to the presence of stearic acid as the main component of FSHO (~ % 87). The blend of soybean oil and FSHO (1:1) also showed one single peak at 43.09 °C. The combination of liquid and solid lipids is important to keep the stability and structural characterization of NLC. Ref. [54] employed DSC to recognize any variation of crystalline properties of lipid-based carriers. In the propolis wax-based NLC formulations, broad endotherms were related to the heterogeneous composition of this compound. A mixture of behenate glycerol and beeswax did not show a melting peak at 74.5 °C; it was related to structural modification by mixing these two compounds. They reported a reduction of the melting point by the addition of liquid lipid (oil) due to the formation of a lower-ordered crystalline structure and finally more entrapment of encapsulated molecules. Ref. [39] observed sharp endothermic peaks for hesperetin and pure lipids, which indicate their crystalline structure. The hesperetin-loaded NLC showed a shallow and broad endothermic peak and a lower melting point, which indicates a less-ordered crystalline structure. DSC analysis of the astaxanthin oleoresin-loaded NLCs showed a single endothermic peak that had a wider width, lower height, and lower melting point than the pure NLC, which could be attributed to the presence of oleoresin and, hence, less ordered crystalline structure [16].

### 3.5. XRD Analysis

XRD analysis is a useful test for evaluating the (1) crystalline form of NLC (polymorphism) and (2) how the active substance spreads and distributes inside the lipid-based carrier. To evaluate the crystallinity of the resulting NLC systems, the XRD analysis of the pure sage extract (SE) powder, pure NLC (without SE and GL), gelatin-coated NLC without SE, and SE-loaded GL-coated NLC (optimum sample) were performed (Figure 5). According to Figure 6, both the pure NLC samples and GL-coated NLC without SE showed the same crystal patterns with different peak intensities. In the diffractogram of the pure NLC, peaks could be observed at the scattering angle (2θ) ~20.32° (d spacing = 0.44 nm) and 22.21° (d spacing = 0.40 nm), which showed the presence of intermediate β-β′ and sub α polymorphs. It is noted that the d-spacing values of lipid crystals are often the following: α = 0.410, β′ = 0.426 (major peak), 0.385 and β = 0.457 (major peak), 0.390, 0.366 nm. The XRD pattern of GL-coated NLC without SE indicated peaks at 2θ = 19.36° (d spacing = 0.46) and 22.98° (d spacing = 0.39), which shows the presence of β crystals. Therefore, it could be concluded that the gelatin coating changed the crystalline structure of the NLC toward a more ordered form. The polymorphism phenomenon in lipid materials is influenced by internal factors such as (molecular configuration and the presence of impurities) and external factors such as (temperature, pressure, cooling rate, etc.). Polymorphism can lead to the formation of more stable perfect β crystals, which have low space in their lattice to accommodate active compounds, which in turn decreases encapsulation stability during storage time. On the other hand, it increases the surface and hydrophobic interactions and, subsequently, increases the colloidal instabilities. 

In the XRD graph of sage extract powder, diffraction peaks were observed at 2θ = 16°, 18°, 19°, and 22°, indicating that the extract powder had a semicrystalline nature with larger amorphous regions, which could be attributed to the crystalline nature of different ingredients, such as some polyphenolic compounds, salts and sugars. The intensity of peaks in the diffractogram of the optimum NLCs were higher than those without SE, indicating an increase in crystallinity due to the incorporation of the extract. Ref. [56] evaluated the optimum phytosterol-loaded NLC formulations (based on the lowest particle size and PDI) by X-ray diffraction. They reported the presence of both β and β’ crystals, which shows polymorphic transition. They declared that the presence of span 60 emulsifier facilitated the polymorphic transition and so, only the characteristic peak of β crystal was observed. They also reported the presence of only β crystal in dried optimum NLC and NLC–phytosterol formulations. Ref. [16] extracted astaxanthin oleoresin from the alga *Haematococcus pluvialis* and encapsulated it into NLC. The lower intensity of peaks in the X-ray diffraction of the loaded NLCs showed a less ordered crystalline structure. Ref. [54] studied X-ray diffraction of the pomegranate seed oil-loaded NLC and SLN carriers and reported that the intensity of the main diffraction peaks of NLCs were lower than SLN diffraction peaks. Lower intensity is a sign of a higher proportion of β’ crystals and thus, the formation of a lower degree of crystallinity, which in turn leads to higher encapsulation efficiency in NLC.

### 3.6. Antioxidant Activity

Antioxidant capacity measurements of free extract, NLCs without SE, and optimum NLCs immediately after production and 30 days after production were performed with DPPH radical scavenging activity (Figure 7). Immediately after production (first day), the free extract and optimum NLCs showed antioxidant activity close together. However, after 30 days of storage, the antioxidant activity of the free extract was lower than optimum NLCs. This clearly showed the protective effect of the NLC carrier on the antioxidant activity of SE, indicating a more effective antioxidant agent for increasing foods’ shelf life [65]. Similar findings have also been observed in previous studies that used different formulations for bioactive compound encapsulation to increase their antioxidant activity [66,67,68,69]. It should be noted that the antioxidant activity of SE was related to the presence of several compounds with high antioxidant activity, such as rosmarinic acid, carnosic acid, salvianolic acid, and tannins (salvia tannin) [70]. These compounds have the ability to supply hydrogen to stabilize radicals [71]. In addition, the antioxidant properties of NLC without extract were also investigated. This sample contains rosemary essential oil (REO) as a component of liquid lipid in its structure. Based on previous studies, it has been confirmed that REO has high antioxidant activity [72,73,74]. However, our study showed that the antioxidant activity of the extract is higher than REO. Ref. [75] enriched butter samples with free beta-sitosterol and beta-sitosterol-encapsulated NLC; the antioxidant activity of both samples was then measured during three months of storage. Results showed that the decrement of antioxidant activity of butter samples containing free beta-sitosterol was more than those containing the NLC carrier, showing the suitable protective effect of NLC on beta-sitosterol. Ref. [18] confirmed higher DPPH antioxidant activity of turmeric extract–NLC in comparison to free turmeric extract. The antioxidant activity of turmeric acid is related to the presence of curcuminoids and hydroxyl groups in its composition. They reported that antioxidant activity decreased in both samples but the level of decrease was higher in the free turmeric extract than the encapsulated one. 

### 3.7. MIC, and MBC Analysis

The results related to the MIC and the MBC tests for each free extract, free REO, and optimum NLC with and without SE are shown in Table 5. According to the obtained results, the MIC of free extract, free REO, and optimum NLC with and without SE against *S. aureus*, *E. coli*, and *P. aeruginosa* were approximately 0.2, 0.2–0.3, 0.1–0.2, and 0.2–0.3 mg/mL, respectively. The optimum-loaded NLC sample showed the strongest inhibitory effect against *E. coli* and *P. aeruginosa* according to MIC value, while optimum NLC without extract had higher MIC values against *S. aureus* and *E. coli*. The optimum extract-loaded NLC sample also had the strongest antimicrobial properties against all studied micro-organisms based on MBC value. Free rosemary essential oil showed the lowest MBC against *E. coli* and *P. aeruginosa*. Results indicated that free SE and free REO had similar effects on *S. aureus* and *P. aeruginosa* in the MIC test. It should be noted that the obtained results illustrated that optimum NLC without extract had similar effects on pathogenic bacteria in MIC and MBC tests. The result of the current study indicated that optimum SE-loaded NLC showed the strongest antibacterial effects against *E.coli* and *P. aeruginosa*, which could be attributed to having both sage extract and rosemary essential oil in its formulation in the MIC method. The reason for this phenomenon can be similar to the obtained antioxidant results related to the fact that the SE has stronger antioxidant and antibacterial properties than the REO. 

Different antimicrobial mechanisms are suggested for essential oils, such as (1) leaching out cell content by destruction and increasing the permeability of cellular membranes, (2) releasing proton of hydroxyl groups that cause reduced proton gradient and hence, depletion of energy pols in microbial cells, (3) reaction with nucleophilic compounds in microorganism’s structure by electrophilic compounds such as carbonyl groups, (4) inhibition of the synthesis of enzymes in mitochondria and ribosome, and (5) reaction with respiratory enzymes in the membrane. Ref. [76] encapsulated citral into an NLC system and declared lower amounts of MIC and MBC, for encapsulated citral in comparison to its emulsion state. Ref. [18] reported a higher antibacterial effect for the turmeric extract-loaded NLC compared to the free extract NLC against gram-negative and rod-shaped bacteria. They related it to the negative surface charge of particles and the ability of turmeric to enter the membrane of bacteria due to its amphipathic and lipophilic nature.

### 3.8. NLC-Enriched Beef Burger Analysis

#### 3.8.1. Chemical Analysis of Beef Burger

A beef burger is a perishable meat product that could deteriorate easily even under refrigeration conditions. In the present work, the surface pH of beef burger samples treated with free extract and formulated optimum NLC with and without extract was measured on the first day of production, 7 and 90 days after production at two different temperatures (4, and −18 °C) (Figure 8A). The pH of the samples at all three times and different temperatures was around 5.8 and almost constant. In fact, the effects of different treatments on pH were not significantly different. It should also be mentioned that the positive control sample that contained ascorbic acid and sorbate also showed a pH value similar to the treated samples all three times and at different temperatures. However, in the negative control sample, which was the beef burger sample without any additives, an increase in pH (~6) was observed. In the literature, pH values of 6.3 and 6.5 have been reported as the initiation of meat spoilage [77,78]. Oxidation of lipids is one of the important parameters related to the reduction of food quality [50]. The lipid oxidation was determined by measurement of PV and TBARS on the first day of production, and at 7 and 90 days after production at two different temperatures (4, and −18 °C). The PV of beef burger samples treated with free extract-formulated optimum NLC with and without extract during the mentioned period is illustrated in Figure 8B. No significant difference (*p* ≤ 0.05) was observed between the PV of samples immediately after production. The highest increment of PV value was achieved in the control sample and free extract, while the minimum increment was in the positive control sample during 90 days of storage. Additionally, the PV value increased in the free extract more than the optimum NLC formulation during 90 days of storage. The PV of the negative-control sample was significantly enhanced (*p* < 0.05) and its maximum value was obtained on day 90 (2.89 ± 0.08 meq O_2_/kg sample). The PV of all the treated beef burger samples enhanced with time, but at a slower rate in comparison with the negative-control sample. The present study proved that the formulated NLC-containing extract could effectively reduce the lipid oxidation of beef burger samples. The reason for this phenomenon is related to the protective effect of NLC systems on the bioavailability, controlled release, and antioxidant activity of SE, thus helping to increase the antioxidant potential and decrease the rate of lipid oxidation. The degree of hydrolysis, presence of antioxidants, and oxygen availability are three major factors affecting the auto-oxidation rate of lipids. Therefore, it could be concluded that the decrease in hydrolysis as a result of microbial growth inhibition and the antioxidant activity of natural extract/essential oil are the reasons for decreased PV in the treated sample [79]. These results are in good agreement with previous studies that reported that encapsulating bioactive compounds improve their physicochemical performance [19,80,81]. Hydroperoxides are the primary products of auto-oxidation, which in themselves are odorless. However, their decomposition to secondary products leads to the formation of a broad range of carbonyl compounds (such as Malondialdehyde), hydrocarbons, furans, and other products that contributes to the rancid taste of decaying food. Thiobarbituric acid reactive substances (TBARS) are formed as a byproduct of lipid peroxidation, and the TBARS value is widely used as an indicator of the degree of lipid oxidation in foods. Malondialdehyde (MDA) is one of several end products formed through the decomposition of lipid peroxidation products. MDA is a highly reactive three-carbon dialdehyde produced as a byproduct of polyunsaturated fatty acid peroxidation and reacts with TBARS reagent. The increase in TBARS value during storage is due to the increased oxidation of unsaturated fatty acids. The bitterness of meat, loss of color, and unpleasant sensorial characteristics increase as the TBARS values increase [44,78,82,83,84]. The results of TBARS of beef burger samples treated with free extract-formulated optimum NLC with and without extract are given in Figure 8C. The TBARS values of the negative-control sample on the day of production, 7 (4 °C), and 90 (−18 °C) days after production were 0.044 ± 0.004, 1.86 ± 0.169, and 3.143 ± 0.21 mg of MDA/kg of the sample, respectively. The TBARS of all the treated beef burger samples enhanced with time, but at a slower rate in comparison with the negative-control sample. Although all three agents significantly inhibited lipid oxidation and reduced TBARS, the greatest reduction effect was related to the optimum NLC with extract, so that on day 90, the TBARS of the sample treated with it reached 0.15 ± 0.03 mg of MDA/kg of sample. The greatest increase in TBARS was in the control sample, and the minimum increase in this parameter was in the positive control sample and then in the optimum sample during 90 days of storage. Also, TBARS was higher in free extract than in optimum NLC formulation during 90 days of storage. Ref. [54] evaluated peroxide value, as an indicator of chemical stability, in fresh NLC formulations (ranging from 3.51 to 5.07 meq/kg) and showed a slight increment after 40 days of storage. This low change in peroxide value could be related to the protective effect of NLC on the core lipid phase and prevents it from reacting with oxygen. In another research study, rosemary extract, in free and lipid carrier-encapsulated forms, could delay lipid oxidation in beef burgers and extend its shelf life up to 21 days and the encapsulated form showed a greater antioxidant effect than the free form [85].

#### 3.8.2. Microbial Analysis of Beef Burger

The changes in the total counts of *mesophilic bacteria*, *psychotropic bacteria*, *S. aureus*, *coliform*, *E. coli*, molds, and yeasts of treated beef burger samples during 0, 3, and 7 days’ storage are presented in Table 6. The treated beef burger samples with optimum NLC showed a significantly (*p* ˂ 0.05) lower increase in the aerobic mesophilic bacteria population during storage compared to the control sample (beef burger sample without any additives); thus, on days 0, 3, and 7, the population of aerobic mesophilic bacteria in the control sample was 4.29 ± 0.04, 6.86 ± 0.05, and 8.05 ± 0.04 Log CFU/g, respectively, and in the treated beef burger samples with optimum NLC, it had Log CFU/g of 4.29 ± 0.03, 4.40 ± 0.03, and 4.48 ± 0.03, respectively Aerobic mesophilic bacteria are used as an important microbiological indicator to evaluate the health of the production process, the safety evaluation, and as an indicator of spoilage of raw meat [86]. The International Commission on Microbiological Specifications for Foods (ICMSF) maximum allowable amount of total mesophilic bacteria in meat at refrigerated temperature is between 10^6^ and 10^7^ CFU/g (equivalent to 7–6 Log CFU/g) [87]. On the first day, the population of aerobic mesophilic bacteria was around 4.29 Log CFU/g, which indicates the very good quality of the beef burger sample. During the storage period, the total number of aerobic mesophilic bacteria in the treated and control groups increased steadily, but this increase was lower in the treated samples. The antibacterial activity of SE was related to the attendance of several phenolic compounds such as rosmarinic acid, carnosic acid, salvianolic acid, and tannins (salvia tannin), which can have positive effects on inhibiting the growth of bacteria [70,88]. Moreover, since the highest antibacterial activity observed was related to the treated sample with the optimum NLC, it confirms the protective effect of NLC systems on SE bioactivity. The population of other tested bacteria, similar to mesophilic bacteria, showed an increasing trend during storage, and this increase was significantly less in the treated samples with optimum NLC than in other samples. Ref. [21] reported the highest antibacterial effect of *Mentha pulegium* essential oil-loaded NLC was against gram-positive bacteria of *Staphylococcus epidermidis*, *Staphylococcus aureus*, and *Listeria monocytogenes* and also negative bacteria of *Escherichia coli* and *Pseudomonas aeruginosa*. Ref. [76] declared higher antimicrobial activity for citral-loaded NLC in comparison to the free form of citral.

#### 3.8.3. Sensory Evaluation of Beef Burger

Important sensory changes in the color, smell, taste, and texture of meats occur during storage, which are caused by bacterial growth and chemical changes such as oxidation, proteolysis, and production of volatile compounds. These unpleasant changes reduce the shelf life and acceptability of meat products. The effect of the free extract, formulated optimum NLC with and without extract on the sensory properties of beef burger samples on the first day of production, the 7th and 90th days of storage is shown in Figure 8D. No significant difference (*p* ≤ 0.05) was observed between the sensory attribute scores of beef burger samples immediately after production. However, over time, the highest acceptability and acceptance were reported for samples treated with optimized nanostructure and positive control. These better sensory properties in the treated beef burger samples could be attributed to lower proteolytic and lipolytic reactions and lower microbial activities. The greatest decrease in the sensory score was in the control sample during 90 days of storage. Generally, when the population of microorganisms exceeds 7 log CFU/g, off odors can be smelled, attributable to the protein breakdown and accumulation of nonprotein compounds [31]. Ref. [25] reported no significant difference between the sensory properties of the control sample and the fortified sauce sample enriched by the Spanish olive leaf extract-loaded nanostructured lipid carriers. Ref. [39] fortified milk with the hesperetin-loaded NLC. Their results showed a successful bitter-taste masking capacity of NLC. Also, encapsulation did not affect the color properties of milk.

## 4. Conclusions

In the present work, sage extract-loaded/gelatin-coated NLCs have been successfully produced with food-grade lipids including tallow oil and rosemary essential oil. The optimization of the formulation based on particle size and encapsulation efficiency resulted in an optimum NLC sample with a low particle size (100.4 nm), low PDI (0.36), high negative zeta potential (−18.4 mV), and high EE (~80%). The antioxidant, and antimicrobial activity of sage extract as encapsulated in gelatin-coated NLC was significantly higher than free extract. This indicates an increase in the bioactivity stability of the extract by encapsulation and also the antioxidant and antimicrobial properties of NLC itself due to the presence of rosemary in its formulation. Treatment of beef burgers with synthesized optimum NLC containing SE had a significant effect on the shelf-life and quality parameters extension with a considerable reduction of PV, TBARS, and total count of tested microorganisms. These findings illustrate the potential efficiency of this produced antioxidant and antimicrobial NLC system as an effective technique and natural preservative for sustaining quality and extending the shelf life of beef burger samples.

## Figures and Tables

**Figure 1 foods-12-03737-f001:**
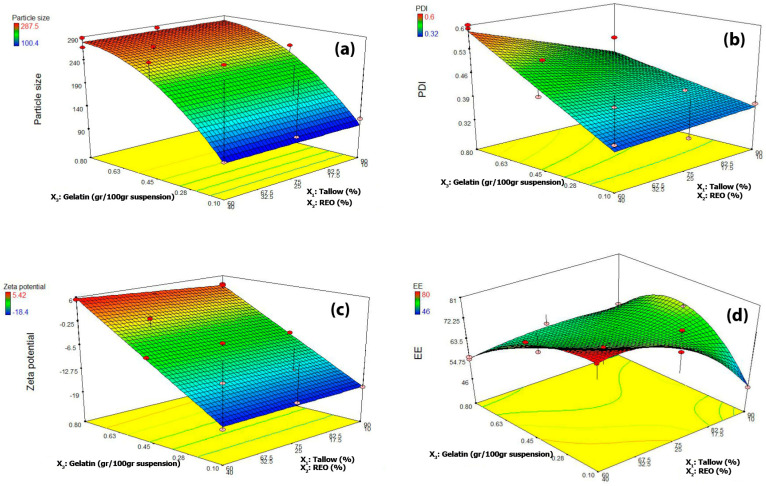
Perturbation plots showing the effect of independent factors on the responses (**a**) Particle size, (**b**) Polydispersity index, (**c**) Zeta Potential, and (**d**) Encapsulation efficiency. Where A is the solid lipid (*w*/*w* %), B is the liquid lipid (*w*/*w* %), and C is the gelatin concentration (%).

**Figure 2 foods-12-03737-f002:**
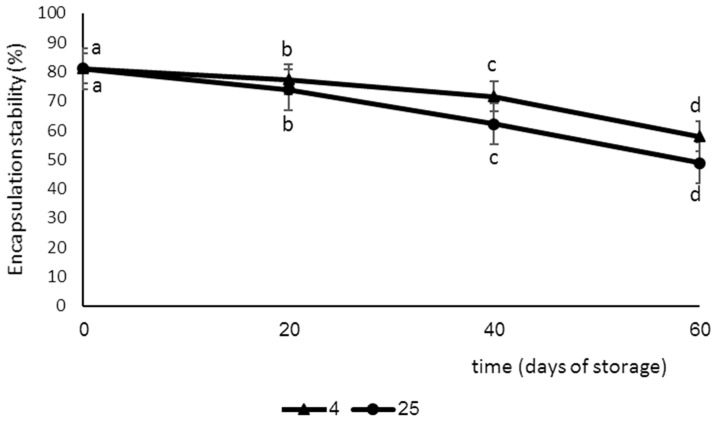
The encapsulation stability (ES%) of prepared sage extract-loaded nanostructured lipid carriers (SE-NLC) for 60 days at 4 °C and 25 °C. Data are expressed as mean ± SD (*n* = 3). Bars in the figure without the same superscripts (a–d) represent significant differences at the 0.05 level according to the Duncan test between encapsulation stability in the different day (*p* < 0.05).

**Figure 3 foods-12-03737-f003:**
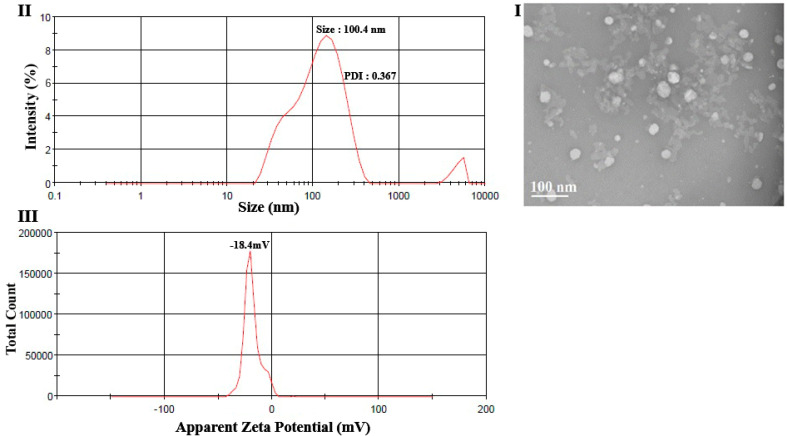
TEM image (**I**), particle size (**II**), and zeta potential (**III**)of the optimized NLC.

**Figure 4 foods-12-03737-f004:**
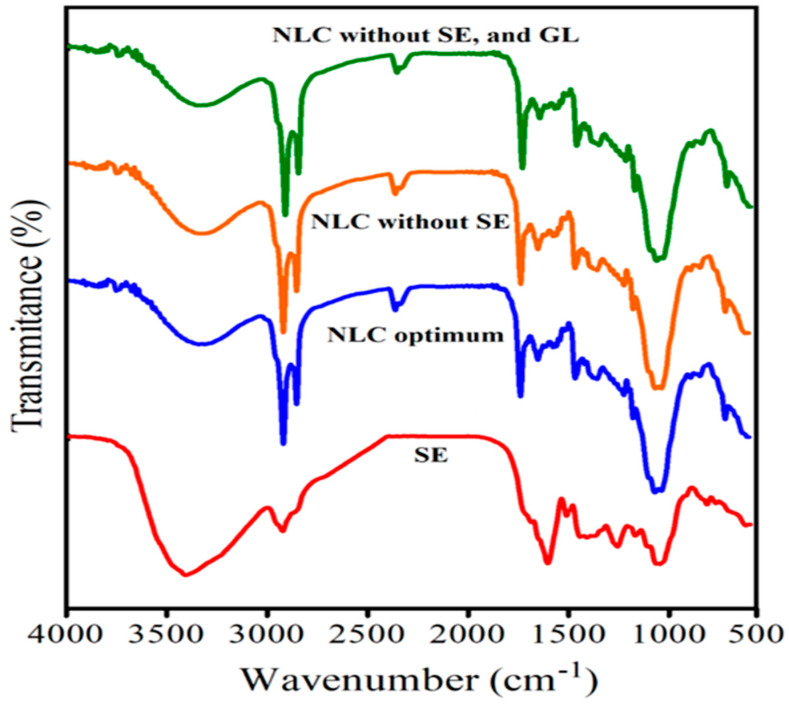
Fourier transform infrared (FTIR) spectra of sage extract (SE), gelatin (GL), nanocarrier structure lipid without SE, and GL (NLC without SE, and GL), GL-coated NLC, without SE, and SE-loaded GL-coated NLC (optimum NLC).

**Figure 5 foods-12-03737-f005:**
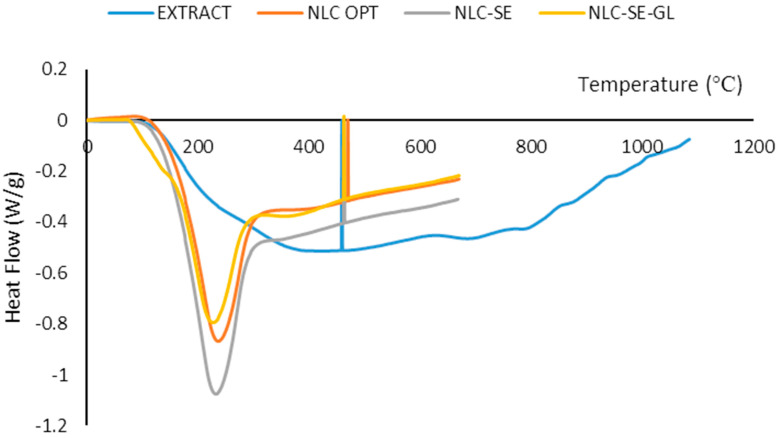
Differential scanning calorimetry (DSC) of free sage extract, sage extract-loaded gelatin-coated nanocarrier structure lipid (NLC OPT), NLC without SE (NLC-SE), and NLC-SE without coated gelatin (NLC-SE-GL).

**Figure 6 foods-12-03737-f006:**
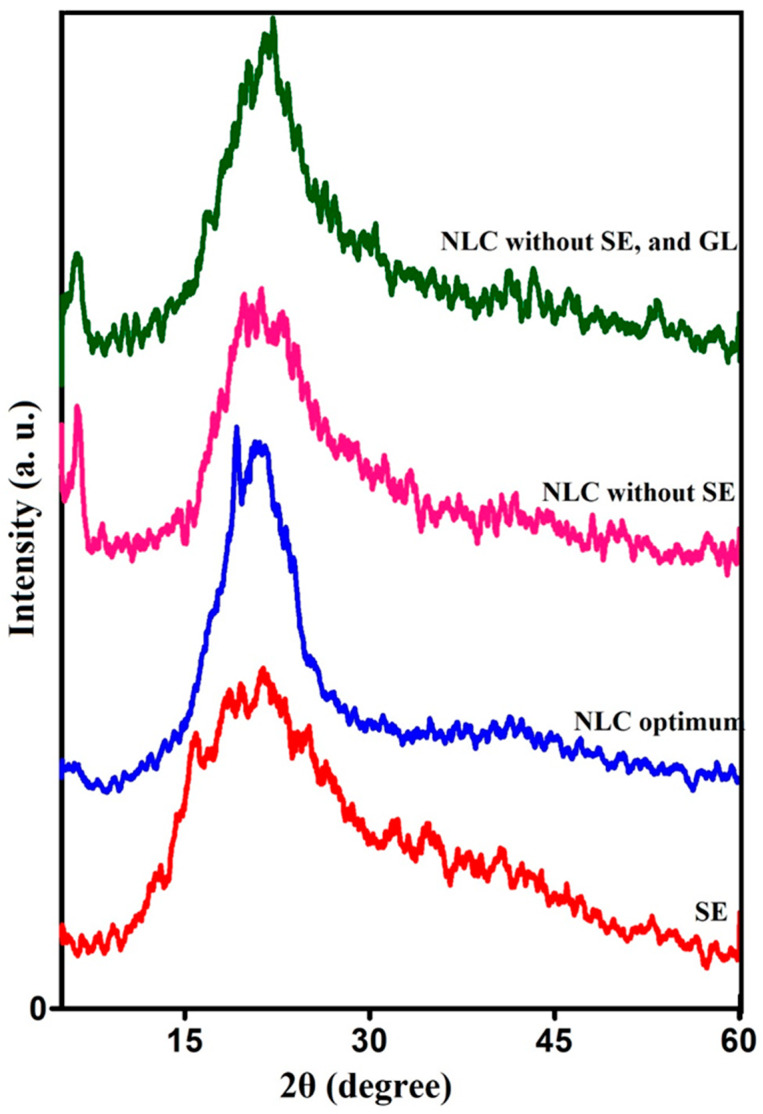
The X-ray diffraction (XRD) pattern of free sage extract (SE), SE-loaded gelatin (GL)-coated nanocarrier structure lipid (NLC optimum), NLC without SE, NLC without SE, and GL.

**Figure 7 foods-12-03737-f007:**
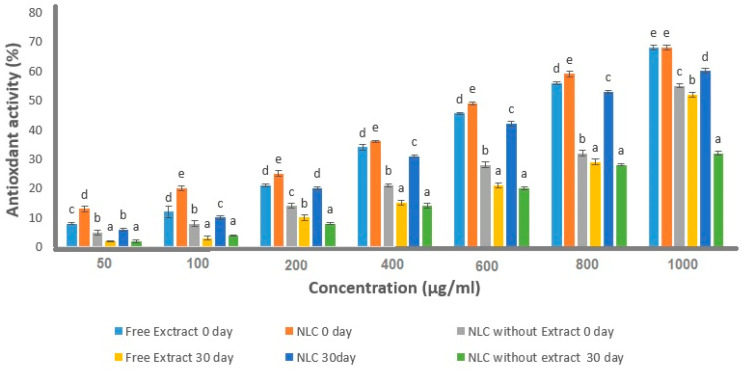
Antioxidant activity of free sage extract (SE) on day 0, optimum-formulated nanocarrier structure lipid (NLC) on day 0, NLC without extract on day 0, free sage extract (SE) on day 30, optimum-formulated nanocarrier structure lipid (NLC) on day 30, and NLC without extract on day 30. Bars in the figure without the same superscripts (a–e) represent significant differences at the 0.05 level according to the Duncan test between antioxidant activity at different form samples (*p*
< 0.05).

**Figure 8 foods-12-03737-f008:**
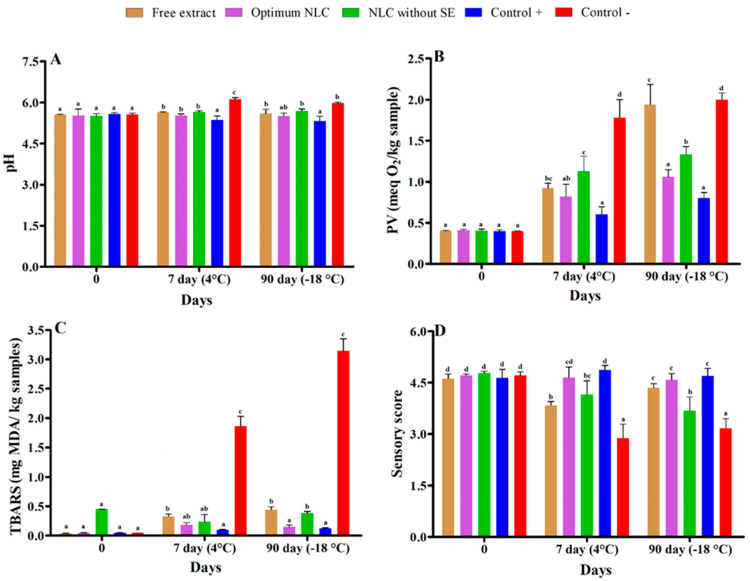
The pH (**A**), peroxide value (PV) (**B**), thiobarbituric acid reactive substances (TBARS) (**C**), and sensory score (**D**) of beef burger samples treated with free extract, optimum-formulated nanocarrier structure lipid (NLC) with and without extract immediately after production (0), 7 (4 °C), and 90 (−18 °C) days after production. Superscripts (a–d) show significant differences between groups. Bars in the figure without the same superscripts (a and d) represent significant differences at the 0.05 level according to the Duncan test between pH, PV, TBARS, and sensory score at different forms of NLC (*p*
< 0.05).

**Table 1 foods-12-03737-t001:** Types of beef burgers produced, and their ingredients.

Samples	Salt (%)	Spice (%)	Water (%)	Free SE (g/100 g)	SE-Loaded NLC (g/100 g)	NLC without SE (g/100 g)	Ascorbic Acid (mg/kg)	Sorbate (mg/kg)
1	1.5	2	1.5	0.2	-	-	-	-
2	1.5	2	1.5	-	0.2	-	-	-
3	1.5	2	1.5	-	-	0.3	-	-
4 (positive control)	1.5	2	1.5	-	-	-	300	200, 1500
5 (control)	1.5	2	1.5	-	-	-	-	-

**Table 2 foods-12-03737-t002:** Combined D-optimal design for independent, and dependent factors and their Output data.

Run No.	Solid Lipid (*w*/*w* %) (*X*_1_)	Liquid Lipid (*w*/*w* %) (*X*_2_)	Gelatin Concentration (g/100 g Suspension) (*X*_3_)	Particle Size (nm)	PDI	ZP (mV)	EE (%)
1	60	40	0.10	100.4	0.36	−18.4	80
2	90	10	0.80	273.3	0.49	3.5	56
3	60	40	0.80	275.5	0.59	5.42	55
4	82.5	17.5	0.28	169.23	0.41	−13.2	65
5	67.5	32.5	0.28	132.7	0.41	−11.98	74
6	60	40	0.80	287.5	0.60	5.4	56
7	75	25	0.45	231	0.42	−5.91	63
8	75	25	0.80	284.6	0.48	3.48	60
9	90	10	0.80	272.9	0.42	3.1	56
10	90	10	0.1	112.52	0.37	−17.6	46
11	67.5	32.5	0.63	269.2	0.5	0.54	61
12	90	10	0.45	236	0.39	−6.8	69
13	75	25	0.1	100.2	0.32	−17	73
14	60	40	0.45	234.3	0.44	−5.51	75

**Table 3 foods-12-03737-t003:** Regression coefficients for the response variables and analysis of variance of the regression models.

Source	Particle Size		PDI		ZP		EE	
	F-Value	*p*-Value	F-Value	*p*-Value	F-Value	*p*-Value	F-Value	*p*-Value
*X* _3_	345.73	<0.0001 ***	-	-	-	-	-	-
*X* ^2^ _3_	17.22	0.0016 **	-	-	-	-	-	-
*X* _1_ *X* _2_	-	-	-	-	-	-	54.73	0.0007 ***
*X* _1_ *X* _3_	-	-	8.59	0.0150 *	711.75	<0.0001 ***	68.24	0.0004 ***
*X* _2_ *X* _3_	-	-	70.91	<0.0001 ***	918.16	<0.0001 ***	423.42	<0.0001 ***
*X* _1_ *X* _2_ *X* _3_	-	-	-	-	-	-	18.27	0.0079 **
*X* _1_ *X* ^2^ _3_	-	-	-	-	-	-	249.58	<0.0001 ***
*X* _2_ *X* ^2^ _3_	-	-	-	-	-	-	30.12	0.0027 **
*X* _1_ *X* _2_ *X* ^2^ _3_	-	-	-	-	-	-	118.12	0.0001 ***
Model	173.02	<0.0001 ***	37.44	<0.0001 ***	679.00	<0.0001 **	147.83	<0.0001 ***
Linear mixture	-	-	19.68	0.0013 **	15.14	0.0030 **	196.23	<0.0001 ***
LOF	6.42	0.14	0.46	0.82	15.86	0.06	5.97	0.15

Values are represented as mean ± SD (*n* = 3), * *p*, ** *p*, and *** *p* indicated significant differences at the 0.05 level.

**Table 4 foods-12-03737-t004:** The suggested fitted models for different responses and model summary statistics in combined D optimal design for the NLC formulations.

Source	Suggested Models		Sequential *p*-Value		Partial Sum of Squares		Lack of Fit (LOF)	Model Summary Statistics (MSS)		
	Mix Order	Process Order	Mix	Process	Sum of Squares	Mean Square		R^2^	Adj-R^2^	Pred-R^2^
Particle Size	Mean	Quadratic	-	0.0016 **	67,857.38	33,928.69	0.14	0.97	0.96	0.95
PDI	Linear	Linear	0.0009 ***	<0.0001 ***	0.079	0.026	0.82	0.92	0.89	0.81
ZP	Linear	Linear	0.0153 *	<0.0001 ***	1052.88	350.96	0.06	0.99	0.99	0.99
EE	Quadratic	Quadratic	0.0005 ***	<0.0001 ***	1176.45	147.06	0.15	0.99	0.99	0.83

*, **, *** Significant at *p*-level ˂ 0.05, *p*-level ˂ 0.01, *p*-level ˂ 0.001, respectively.

**Table 5 foods-12-03737-t005:** The data of minimum inhibitory concentration (MIC) and minimum bactericidal concentration (MBC) for free extract, free rosemary essential oil (REO), and optimum-formulated NLC with and without extract.

Bacteria			MIC (mg/mL)	
	Free Extract	Free REO	Optimum NLC	Optimum NLC without Extract
*S. aureus*	0.2 ± 0.00	0.2 ± 0.00	0.2 ± 0.00	0.3 ± 0.00
*E. coli*	0.2 ± 0.00	0.3 ± 0.00	0.1 ± 0.00	0.3 ± 0.00
*P. aeruginosa*	0.2 ± 0.00	0.2 ± 0.00	0.1 ± 0.00	0.2 ± 0.00
			MBC (mg/mL)	
*S. aureus*	0.3 ± 0.00	0.3 ± 0.00	0.1 ± 0.00	0.3 ± 0.00
*E. coli*	0.3 ± 0.00	0.4 ± 0.00	0.1 ± 0.00	0.3 ± 0.00
*P. aeruginosa*	0.2 ± 0.00	0.3 ± 0.00	0.2 ± 0.00	0.2 ± 0.00

Data are expressed as mean ± SD (*n* = 3).

**Table 6 foods-12-03737-t006:** Microbial analysis of beef burger samples treated with Free extract (FE), optimum-formulated NLC, and NLC without extract (NLC-E) during 0, 3, and 7 days’ storage.

Microbial Count (log CFU/g)	Group	Storage Day		
		0	3	7
Aerobic mesophilic bacteria	FE	4.29 ± 0.03 ^a^	4.69 ± 0.02 ^a^	5.06 ± 0.06 ^d^
	NLC	4.29 ± 0.03 ^a^	4.40 ± 0.03 ^a^	4.48 ± 0.03 ^b^
	NLC-E	4.28 ± 0.03 ^a^	4.47 ± 0.02 ^a^	4.69 ± 0.005 ^c^
	C_P_	4.26 ± 0.08 ^a^	4.07 ± 0.18 ^a^	3.89 ± 0.007 ^a^
	C^N^	4.29 ± 0.04 ^a^	6.86 ± 0.05 ^b^	8.05 ± 0.04 ^e^
*Staphylococcus aureus*	FE	0.93 ± 0.02 ^a^	2.30 ± 0.02 ^d^	2.57 ± 0.06 ^d^
	NLC	0.91 ± 0.02 ^a^	2.03 ± 0.03 ^b^	2.08 ± 0.02 ^b^
	NLC-E	0.92 ± 0.02 ^a^	2.17 ± 0.02 ^c^	2.32 ± 0.02 ^c^
	C_P_	0.9 ± 0.005 ^a^	0.98 ± 0.03 ^a^	0.96 ± 0.03 ^a^
	C^N^	0.92 ± 0.02 ^a^	3.07 ± 0.01 ^e^	3.34 ± 0.01 ^e^
Coliforms	FE	2.17 ± 0.04 ^a^	2.58 ± 0.02 ^c^	3.47 ± 0.02 ^c^
	NLC	2.16 ± 0.04 ^a^	2.47 ± 0.03 ^b^	3.34 ± 0.04 ^b^
	NLC-E	2.15 ± 0.02 ^a^	2.60 ± 0.004 ^c^	3.48 ± 0.02 ^c^
	C_P_	2.15 ± 0.04 ^a^	2.10 ± 0.03 ^a^	2.07 ± 0.04 ^a^
	C^N^	2.16 ± 0.02 ^a^	3.47 ± 0.03 ^d^	3.76 ± 0.005 ^d^
Yeasts and molds	FE	2.29 ± 0.08 ^a^	2.69 ± 0.01 ^d^	3.03 ± 0.02 ^d^
	NLC	2.32 ± 0.03 ^a^	2.38 ± 0.03 ^b^	2.72 ± 0.02 ^b^
	NLC-E	2.33 ± 0.04 ^a^	2.50 ± 0.04 ^c^	2.78 ± 0.01^b^ ^c^
	C_P_	2.31 ± 0.04 ^a^	2.18 ± 0.02 ^a^	2.09 ± 0.03 ^a^
	C^N^	2.32 ± 0.05 ^a^	3.08 ± 0.04 ^e^	3.48 ± 0.04 ^e^
Psychotropic bacteria	FE	2.29 ± 0.04 ^a^	2.54 ± 0.02 ^d^	2.63 ± 0.02 ^d^
	NLC	2.22 ± 0.04 ^a^	2.39 ± 0.01 ^b^	2.47 ± 0.01 ^b^
	NLC-E	2.30 ± 0.05 ^a^	2.47 ± 0.01 ^c^	2.6 ± 0.01 ^c^
	C_P_	2.29 ± 0.05 ^a^	2.22 ± 0.02 ^a^	2.17 ± 0.04 ^a^
	C^N^	2.25 ± 0.05 ^a^	3.30 ± 0.03 ^e^	3.58 ± 0.02 ^e^
*E. coli*	FE	1.81 ± 0.017 ^a^	1.93 ± 0.11 ^b^	2.15 ± 0.07 ^c^
	NLC	1.8 ± 0.005 ^a^	1.79 ± 0.13 ^b^	1.92 ± 0.04 ^b^
	NLC-E	1.8 ± 0.007 ^a^	2.00 ± 0.11 ^b^	2.22 ± 0.08 ^c^
	C_P_	1.8 ± 0.005 ^a^	1.54 ± 0.15 ^a^	1.44 ± 0.09 ^a^
	C^N^	1.81 ± 0.005 ^a^	2.41 ± 0.08 ^c^	2.78 ± 0.21 ^d^

Positive-control sample (C_P_), and Negative-control sample (C^N^). Superscripts (a–e) show significant differences between groups (*p* < 0.05). Data are expressed as mean ±SD (*n* = 3).

## Data Availability

Data is contained within the article.

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
