# Peer review of "The Gelatin-Coated Nanostructured Lipid Carrier (NLC) Containing Salvia officinalis Extract: Optimization by Combined D-Optimal Design and Its Application to Improve the Quality Parameters of Beef Burger"

_foods, 2023, doi:10.3390/foods12203737_

Round 1

Reviewer 1 Report

1. The authors said that "S. aureus (ATCC25923), P. aeruginosa (ATCC27853), and E. coli (ATCC25922) were 197 cultured on sterile agar plates and incubated for 24 h at 37 °C" Please specify the agar used. 

2. Line 206: Write the names of microorganisms in Italics

3. Line 229: "After preparation, the burgers were placed in sterile con- 228 tainers and stored at 4°C for 7 days and 18°C for 3 months." I believe that the right is -18°C.

4. In Table 5 write the names of microorganisms in Italics

5. Line 267: Specify which culture media was used.

6. Line 271-274: write the names of microorganisms in Italics

7. Line 272: "Samples inoculated with Salmonella were plated on Xylose Lysine Deoxycholate 272 agar (XLD) overlayed with Tetrathionate broth..." What did the authors mean by that? Are there beef burger samples inoculated with Salmonella during their experiments? Please explain. It's not clear. 

8. There is a discrepancy between the two statements regarding the days of analysis. In Line 229, it is stated that "Microbial tests were performed at 0, 3, and 7 days at 4 °C," while in Lines 727-728, it is mentioned that "The changes in the total counts of mesophilic bacteria, psychotropic bacteria, S. aureus, coliform, E. coli, molds, and yeasts of treated beef samples during 1, 3, and 7 days’ storage are presented in Table 6."  Please correct this inconsistency throughout the document. You should make sure the days of analysis match.

Author Response

The authors said that "S. aureus (ATCC25923), P. aeruginosa (ATCC27853), and E. coli (ATCC25922) were 197 cultured on sterile agar plates and incubated for 24 h at 37 °C" Please specify the agar used.

Answer:

They were specified and highlighted in the text

  1. Line 206: Write the names of microorganisms in Italics

Answer:

The correction was made.

  1. Line 229: "After preparation, the burgers were placed in sterile con- 228 tainers and stored at 4°C for 7 days and 18°C for 3 months." I believe that the right is -18°C.

Answer:

Yes, it was corrected

  1. In Table 5 write the names of microorganisms in Italics

Answer:

It was corrected.

  1. Line 267: Specify which culture media was used.

Answer:

The culture media was highlighted in the line 268-274.

  1. Line 271-274: write the names of microorganisms in Italics

Answer:

It was corrected

  1. Line 272: "Samples inoculated with Salmonella were plated on Xylose Lysine Deoxycholate 272 agar (XLD) overlayed with Tetrathionate broth..." What did the authors mean by that? Are there beef burger samples inoculated with Salmonella during their experiments? Please explain. It's not clear.

Answer:

Salmonella was not inoculated in the beef burger samples, but this test was done as a supplementary test, Since the results of the control and others were negative for the Salmonella test, we thought that there was no need to report the negative results in the manuscript. It was corrected.

  1. There is a discrepancy between the two statements regarding the days of analysis. In Line 229, it is stated that "Microbial tests were performed at 0, 3, and 7 days at 4 °C," while in Lines 727-728, it is mentioned that "The changes in the total counts of mesophilic bacteria, psychotropic bacteria, S. aureus, coliform, E. coli, molds, and yeasts of treated beef samples during 1, 3, and 7 days’ storage are presented in Table 6."  Please correct this inconsistency throughout the document. You should make sure the days of analysis match.

Answer:

The first day had been considered as  0 or 1. The correction was made throughout the manuscript.

Reviewer 2 Report

The manuscript entitled ‘The gelatin-coated nanostructured lipid carrier (NLC) containing Salvia officinalis extract: Optimization by combined D-optimal design and its application to improve the quality parameters of beef burger’ details about to gelatin-coated nanostructured lipid carrier to encapsulate sage extract and use this nanoparticle to increase the quality parameters of beef burger samples. As per my view, there are different queries for this work, which should be addressed for further consideration.

·       Authors reported that in the sage extract preparation section “the obtained final extract was freeze-dried at -5 °C with 0.1 mbar”. Why authors used this (-5 °C) for freeze drying and is this temperature ideal for freeze drying and authors obtained a freeze-drying product? Also provide the reference for the same.

·       Authors sometimes mentioned stores in fridge and sometime refrigerated. Need to write it a proper form and it also deviate the storage temperature as well.

·       Authors used the UV spectrophotometer to analyze the SE content in formulation. What is the sensitivity of UV to measure SE extract in terms of LOD and LOQ, and the lambda max give peak for whole extract or any component? Provide the proper reference also.

·       Authors should provide the reference for the encapsulation stability formula provided in the manuscript.

·       Authors performed the FTIR and DSC to find out any incompatibility, they should also justify to perform the XRD study.

·       After preparation, the burgers were placed in sterile containers and stored at 4°C for 7 days and 18°C for 3 months (line no. 228-229). Here it should be -18 °C. please justify the reason to choose -18 °C for the storage along with 4 °C?

·       For the sensory characteristics study, are authors obtained informed consent form all subjects involved in the study? Need to mention in manuscript.

·       Authors need to explain how the change in Zeta potential is responsible for the increase of particle size (line no. 388).

·       How will authors justify the goal for low zeta potential (line no. 422)? As low zeta potential cause instability in formulation and authors goal is to achieve low zeta potential.

·       Need to explain probable reason for this behaviour (level of % ES decrease; line no. 430-431) and % ES decrease was significant or not?

·       Please rewrite the caption for Figure 3 and in Zetasize image, what the second peak shows? As it is not an ideal peak.

·       No standard deviation shown in the Figure no. 7 and authors also need to report statistics in term of significant differences with different samples.

·       Authors should provide SD in Table 5.

In Figure 8 caption and Table 6, authors mentioned Superscripts (a–d) show significant differences between groups. Need to report p value.

There is lack of punctuations/typo errors and a lot of grammatical errors throughout the manuscript. Authors need to improve the English language of overall manuscript. At many points, there is no clarity about the sentences. For example in line no. 63 " has received much attention e in the food field". Line no. "hated at 70 °C for 24 min" it should be heated. 

There are so many such corrections required in whole manuscript.

Author Response

  1. Authors reported that in the sage extract preparation section “the obtained final extract was freeze-dried at -5 °C with 0.1 mbar”. Why authors used this (-5 °C) for freeze drying and is this temperature ideal for freeze drying and authors obtained a freeze-drying product? Also provide the reference for the same.

Answer:

We greatly appreciate your constructive and thoughtful comment. It was modified to: " the obtained final extract was freeze-dried at -40 °C with 0.3 hPa, increasing the temperature of the shelves from −40 °C to 17 °C in 24 and stored at -4 °C for subsequent usage".

The relevant reference was provided.

  1. Authors sometimes mentioned stores in fridge and sometime refrigerated. Need to write it a proper form and it also deviate the storage temperature as well.

Answer:

The produced beef burgers were kept at two temperatures, fridge (-18 °C) and refrigerated (4 °C). Microbial analysis of it treated with Free extract (FE), optimum formulated NLC, and NLC without extract (NLC-E) during 0, 3, and 7 days’ storage in 4 °C. Chemical analysis was checked on 0 and 7 days in 4°C. At the temperature of -18°C for 90 days, only chemical analysis was done in 0 and 90 days.

  1. Authors used the UV spectrophotometer to analyze the SE content in formulation. What is the sensitivity of UV to measure SE extract in terms of LOD and LOQ, and the lambda max give peak for whole extract or any component? Provide the proper reference also.

Answer:

The concentration of the extract was determined based on the maximum absorbance of the extract in the spectrophotometer, and we did not measure the concentration of a certain compound in the extract. It was not known the absorption maximum related to which compound in the extract, and we did not need to know it. This method has been used in various research papers and works. The relevant reference was added.

  1. Authors should provide the reference for the encapsulation stability formula provided in the manuscript.

Answer:

The relevant reference was added.

  1. Authors performed the FTIR and DSC to find out any incompatibility, they should also justify performing the XRD study

Answer:

FTIR and DSC were used for confirmation of entrapment of extract in a carrier. XRD was not necessary for this purpose

  1. After preparation, the burgers were placed in sterile containers and stored at 4°C for 7 days and 18°C for 3 months (line no. 228-229). Here it should be -18 °C. please justify the reason to choose -18 °C for the storage along with 4 °C?

Answer:

In this research, short-term storage ( 7 days in refrigerator at 4 °C), similar to homemade beefburger production conditions, and long-term storage (90 days in fridge at -18 °C), similar to industrial hamburger production conditions, used for evaluating microbial and chemical parameters of beef burger. The produced beef burgers were kept at two temperatures, fridge (-18 °C) and refrigerated (4 °C). Microbial analysis was carried out only for storage at 4 °C because microorganisms have no activity at temperatures below -18 and do not cause spoilage however, chemical changes were analyzed both at 4°C ( for 7 days) and -18°C ( for 90 days).

  1. For the sensory characteristics study, are authors obtained informed consent form all subjects involved in the study? Need to mention in manuscript.

Answer:

We appreciate the review's comment. The correction was made in manuscript.

  1. Authors need to explain how the change in Zeta potential is responsible for the increase of particle size (line no. 388).

Answer:

It was explained. With the increase of zeta potential, the electrostatic repulsion forces between the particles increase and the possibility of particle aggregation and the formation of larger particles decreases. The  NLC systems with zeta potentials higher than ±20 mV potentially have long-term colloidal stability. In this study, with the increase of gelatin concentration, the value of zeta potential changed from – 18.4 to ~ + 8 mV which probably caused instability of the colloidal system, aggregation, and particle size enhancement.

  1. How will authors justify the goal for low zeta potential (line no. 422)? As low zeta potential cause instability in formulation and authors goal is to achieve low zeta potential.

Answer:

Thanks for  your comment. It was corrected. High zeta potential was favorable.

  1. Need to explain probable reason for this behaviour (level of % ES decrease; line no. 430-431) and % ES decrease was significant or not?

Answer:

We appreciate your helpful comment. The following sentence was added: “the ES% level was the same in both temperatures while over time, the level of ES% significantly decreased and this decrease was more at 25 °C than at 4 °C.”

  1. Please rewrite the caption for Figure 3 and in Zetasize image, what the second peak shows? As it is not an ideal peak.

Answer:

The correction was made.

  1. No standard deviation shown in the Figure no. 7 and authors also need to report statistics in term of significant differences with different samples.

Answer:

The correction was made.

13.Authors should provide SD in Table 5.

Answer:

The correction was made.

14.In Figure 8 caption and Table 6, authors mentioned Superscripts (a–d) show significant differences between groups. Need to report p value.

Answer:

The correction was made.

15.Comments on the Quality of English Language

 There is lack of punctuations/typo errors and a lot of grammatical errors throughout the manuscript. Authors need to improve the English language of overall manuscript. At many points, there is no clarity about the sentences. For example in line no. 63 " has received much attention e in the food field". Line no. "hated at 70 °C for 24 min" it should be heated. 

There are so many such corrections required in whole manuscript.

Answer:

Thanks a lot for the valuable comment from the respected reviewer. We have revised the whole manuscript carefully and tried to avoid any spelling or grammar errors.

Round 2

Reviewer 2 Report

Authors addressed all the comments and it may be accepted in current form.